# Towards Effective Emotion Detection: A Comprehensive Machine Learning Approach on EEG Signals

**Ietezaz Ul Hassan** [1,†]**, Raja Hashim Ali** [2,†] **, Zain ul Abideen** [3]**, Ali Zeeshan Ijaz** [2] **and Talha Ali Khan** [4,*]

1. IMaR Research Centre, Munster Technological University, V92 CX88 Tralee, Ireland; letezaz.ul.hassan@research.ittralee.ie
2. AI Research Group, Faculty of Computer Science and Engineering, Ghulam Ishaq Khan Institute of Engineering Sciences and Technology, Topi 23640, Khyber Pakhtunkhwa, Pakistan; hashim.ali@giki.edu.pk (R.H.A.); ali.zeeshan@giki.edu.pk (A.Z.I.)
3. Computational Soft Matter and Biophysics Lab, Basque Center for Materials, Applications and Nanostructures (BCMaterials), Buil. Martina Casiano, Pl. 3 Parque Científico UPV/EHU Barrio Sarriena, 48940 Leioa, Spain; zainul.abideen@bcmaterials.net
4. Department of Business, University of Europe for Applied Sciences, Think Campus, Konrad-Zuse-Ring 11, 14469 Potsdam, Germany
* Correspondence: talhaali.khan@ue-germany.de
† These authors contributed equally to this work.

**Abstract:** Emotion detection assumes a pivotal role in the evaluation of adverse psychological attributes, such as stress, anxiety, and depression. This study undertakes an exploration into the prospective capacities of machine learning to prognosticate individual emotional states, with an innovative integration of electroencephalogram (EEG) signals as a novel informational foundation. By conducting a comprehensive comparative analysis of an array of machine learning methodologies upon the Kaggle Emotion Detection dataset, the research meticulously fine-tunes classifier parameters across various models, including, but not limited, to random forest, decision trees, logistic regression, support vector machines, nearest centroid, and naive Bayes classifiers. Post hyperparameter optimization, the logistic regression algorithm attains a peak accuracy rate of 97%, a proximate performance mirrored by the random forest model. Through an extensive regimen of EEG-based experimentation, the study underscores the profound potential of machine learning paradigms to significantly elevate the precision of emotion detection, thereby catalyzing advancements within the discipline. An ancillary implication resides in early discernment capabilities, rendering this investigation pertinent within the domain of mental health assessments.

**Keywords:** emotion detection; electroencephalogram (EEG) signals; machine learning

## 1. Introduction

The significance of emotion detection and recognition resonates profoundly across an array of disciplines, casting an influential shadow, particularly over the spheres of mental health and security [1]. The capacity to seamlessly and autonomously detect and decipher emotions, eliminating the necessity for direct human involvement, unfurls a path of remarkable efficiency for real-time assessment of human emotional states [2]. This capability is of paramount importance, especially within high-pressure contexts, like interviews and forensic examinations, where swift and accurate emotion detection can yield valuable insights [3]. This operational framework hinges on the bedrock of supervised machine learning techniques, where the prerequisite for success lies in the availability of painstakingly curated and meticulously processed labeled datasets [4]. The training of classifiers, the crux of this approach, is predicated upon this repository of data, providing the foundation upon which accurate and reliable emotion recognition models can be built [5]. While numerical or dimensional data formats remain viable, the intricate

machinery of machine learning algorithms is most adept at processing categorical inputs, as underscored by Liang et al.'s work in the domain of classification [6].

At the same time, among the multifaceted avenues within this realm, the utilization of electroencephalogram (EEG) signals for emotion detection emerges as a captivating prospect [7]. This technology unfurls a panorama of applications that span from precision-driven control mechanisms in activities such as drone piloting and driving to the immersive landscapes of electronic gaming [8]. However, despite its immense potential, the domain of EEG-based emotion detection has encountered limited exploration. Only a handful of studies have delved into this burgeoning arena, often relying on specific datasets, as observed in the seminal work by Merlin et al. [9].

Interestingly, exploring emotion detection through EEG signals reveals a promising new area [10]. For instance, Taheri et al. [11] used EEG data, emotion detection, and machine learning to assess cognitive workload in pilots. This ability to uncover detailed insights from brainwave data opens up fresh understanding of human emotions. This journey goes beyond limits, offering a chance to understand the complexities of human emotions [12]. Investigating EEG signals as a way to detect emotions is not just about technology; it reshapes how we understand human thinking. The intricate pathways of neural activity, often hidden from awareness, become visible through EEG signals, providing a new way to see emotions. This blend of technology and human consciousness highlights how important EEG signals are in uncovering the mysteries of emotion detection. As we navigate the intersection of technology and human understanding, the importance of EEG signals in emotion detection becomes clear. This has far-reaching impacts, going beyond neuroscience to psychology, computer science, and more. The promise of undiscovered insights urges further exploration, pushing us to fully tap into EEG-based emotion detection. In this uncharted territory, today's knowledge could lead to profound advancements, shedding light on the ever-changing landscape of human emotions.

### 1.1. Significance of Electroencephalography (EEG)

Various data types can be harnessed for emotion detection, with a particularly intriguing source being the electrogram that portrays the scalp's electrical activity. This electrical activity demonstrates dynamic variations corresponding to brain function and reactions linked to specific emotional states. A technique employed for tracking this brain electrical activity is termed electroencephalography (EEG), affording a lens into the study of neural dynamics and the intricate workings of the brain [13]. The EEG method captures the electrogram, which effectively represents the aggregate activity of the brain's outermost layer [14]. Electrodes strategically positioned on the scalp record the electrical field emanating from nerve cells, with electrical impulses serving as the primary means of communication within the nervous system. This macroscopic brain activity is pivotal in diagnosing an array of brain disorders, such as brain tumors, epilepsy, stroke, Alzheimer's disease [15], sleep disorders, encephalopathy, herpes encephalitis, and Creutzfeldt–Jakob disease [16]. Furthermore, EEG signals facilitate the exploration of brain activity characteristics across various cognitive scenarios, encompassing the impact of medications, drugs, and fatigue on brain function. Notably, the EEG patterns during cognitive engagement starkly contrast those observed at rest. For instance, EEG can effectively decipher distinct sleep stages and levels of attentiveness, offering insights into learning and memory processes. Researchers also leverage EEG data to comprehend the brain's responses to stimuli and events, elucidating multifaceted brain functions activated during diverse circumstances and tasks. The measurement of EEG signal strength is typically executed through non-invasive electrodes discreetly placed along the scalp. However, in instances where a more invasive approach is needed, such as intracranial EEG, electrodes are directly implanted within the brain. This invasive method, often referred to as electrocorticography, allows for more precise monitoring of brain electrical activity [17]. Thus, the realm of emotion detection and brain research is significantly enriched by the nuanced insights gleaned from EEG data, underscoring its multifaceted applications and potential. EEG-based emotion recognition has

found applications in various domains, including human–computer interaction, medical diagnosis, and the military.

Note that utilizing a multitude of electrodes, often numbering in the hundreds and placed on an individual's scalp to capture brain activity from diverse regions, electroencephalography (EEG) presents two distinct advantages for investigating brain states. The foremost benefit lies in its capacity for highly accurate temporal measurements, a characteristic intrinsic to all electrical recording devices. EEG technology excels in precisely discerning brain activity, offering a temporal resolution of one millisecond or even less [18,19]. This exceptional temporal precision renders EEG an optimal instrument for capturing the swift electrical events transpiring within the brain. Beyond its temporal precision, EEG offers additional merits. The technology not only remains cost-effective and user-friendly but also operates in a non-invasive manner, as the electrodes are affixed to the scalp rather than being surgically implanted within the brain. Consequently, researchers are granted an unhindered vantage point into the intricate workings of a healthy human brain. This approach to brain monitoring provides an accessible and insightful route for studying brain functionality and dynamics.

Nonetheless, a pivotal limitation of EEG recordings resides in its limited spatial resolution. The EEG signals are captured at the scalp and stem from the collective electric fields generated by a multitude of neurons within the cortex. This results in a spatial resolution constrained to approximately one centimeter per electrode. While the utilization of multiple electrodes can indeed highlight areas of heightened electrical activity, the inherent EEG signal does not furnish intricate details about the precise origin of this activity nor does it discern between activities originating from distinct but proximate sites. In recent times, strides have been made in the domain of EEG analysis techniques, facilitating more refined estimations of the signals' origins. These advancements enable researchers to glean improved insights into the sources and nuances of brain activity, offering a step forward in mitigating the spatial resolution limitations of traditional EEG recordings.

Moreover, the clinical utility of EEG holds notable importance in the diagnosis and treatment of patients grappling with epilepsy and depression [20]. In the context of epilepsy, characterized by electric surges emanating from specific brain regions, EEG proves invaluable. By scrutinizing EEG patterns, researchers can ascertain the occurrence of epileptic seizures and discern the specific type of seizure underway. This critical information is essential for healthcare professionals to administer precise and effective care tailored to the patient's condition. Furthermore, EEG can play a pivotal role in pinpointing the focal point of epileptic seizures, contributing to enhanced diagnostic accuracy and informed medical decision-making [21].

At the same time, Emotion AI, alternatively referred to as affective computing, represents a forefront technological innovation that empowers computers and systems to adeptly perceive, interpret, and replicate human emotions and sentiments. This domain lies at the intersection of diverse disciplines, encompassing computer science, psychology, and cognitive science, to forge this remarkable technological advance. While the notion of computers discerning emotions might appear unconventional, research substantiates their capacity to accurately recognize emotions across visual, textual, and auditory inputs. Leveraging emotion AI, enterprises stand to augment their customer service initiatives and bolster decision-making across sectors like sales, marketing, and customer support, thus amplifying their operational effectiveness and customer engagement.

### 1.2. Machine Learning Applications for Emotion Detection from EEG Signals

Several recent works have made nominal contributions to the field of emotion detection from EEG signals [22] and have explored the application and comparison of machine learning classifiers for emotion detection from EEG signals [23]. Recent research has explored using EEG signals to automate emotion recognition systems. These studies reveal that brain signals, though dynamic, can effectively classify emotions. For example, Rania et al. [24] introduce an innovative model that incorporates empirical mode decompo-

sition/intrinsic mode functions (EMD/IMF) and variational mode decomposition (VMD) for signal processing, distinguishing it from conventional methods. By employing entropy and Higuchi's fractal dimension (HFD) for feature extraction and employing classifiers such as naive Bayes, k-NN, CNN, and decision tree, this approach achieves an impressive 95.20% accuracy in classifying emotional states on the DEAP dataset.

Similarly, Haoron et al. [25] discussed the close connection between emotions and human behavior and their manifestation in EEG signals, emphasizing the difficulty in concealing emotional states. The paper outlines the typical steps involved in EEG-based emotion recognition, spanning data acquisition, preprocessing, feature extraction, feature selection, and classification. Furthermore, it reviews existing methods in this domain and evaluates their classification performance, offering valuable insights and a foundational understanding for researchers in the field of EEG-based emotion recognition. The authors compared six different classifiers and achieved comparable results on the DEAP dataset.

More recently, Houssein et al. [26] provided a comprehensive review of EEG-based BCI emotion recognition techniques, encompassing dataset descriptions, emotion elicitation methods, EEG feature extraction, feature selection, machine learning (including k-nearest neighbor, support vector machine, decision tree, artificial neural network, random forest, and naive Bayes), and deep learning approaches (including convolutional and recurrent neural networks with long short-term memory). Additionally, the paper explores EEG rhythms associated with emotions and the brain's emotional processing regions. It concludes by identifying challenges and future research directions in EEG-based human emotion recognition and classification.

## 2. Our Contribution

### 2.1. Gap Analysis

A few gaps in the existing literature that are being addressed by this study include:

**Comprehensive Evaluation of Classifiers:** The study addresses a gap in the literature by providing a comprehensive evaluation of multiple machine learning classifiers for emotion recognition using EEG data. While previous studies might have focused on specific classifiers, as shown earlier, the focus of these studies is on the DEAP dataset, which is a relatively old dataset extracted with older headsets technology. On the other hand, this study systematically compares a wide range of classifiers, shedding light on their relative strengths and weaknesses for this particular task using the latest publicly available dataset.

**Emotion Recognition from EEG Data:** The use of EEG data for emotion recognition is a relatively novel approach. While previous research might have explored emotion recognition using other types of data or techniques, this study contributes to the literature and reinforces the findings of many recent works [24–26] by specifically investigating the feasibility and effectiveness of using EEG data for this purpose.

**Inherent Advantages of Models:** The study delves into a gap in the literature by identifying potential inherent advantages of certain machine learning classifiers in capturing the underlying patterns within EEG data for emotion recognition. This nuanced exploration of model strengths and their implications for accuracy is a novel perspective that extends beyond traditional performance metrics. As shown by Houssein et al. [26] and through other applications of machine learning algorithms, Ensembl-based approaches applied on decision trees like XGBoost and random forest should ideally outperform other machine learning approaches. This study confirms the earlier reported findings.

**Balanced Dataset for Emotion Recognition:** The use of a balanced dataset for emotion recognition is noteworthy. Many studies might rely on imbalanced datasets, which can lead to biased results. This study addresses this gap by utilizing a balanced dataset, thereby providing more accurate results and unbiased data. Although in real life, most scenarios are generally neutral, this dataset has an advantage of representation of equal number of real-world emotion recognition data.

In essence, the study addresses gaps in the literature related to the comprehensive evaluation of classifiers, the utilization of EEG data for emotion recognition, the exploration of model advantages, the importance of balanced datasets, the real-world applicability of emotion recognition systems, and the potential of deep learning methodologies.

## 2.2. Research Questions

The three main research questions being investigated in the current study can be inferred as follows:

1. **Classifier Performance and Generalization:** How do various machine learning classifiers perform in the task of emotion recognition using EEG data? Specifically, what is the accuracy of these classifiers in both training and validation scenarios, and how well do they generalize their predictions to new, unseen data instances?
2. **Relationship Between Model Complexity and Accuracy:** Is there a relationship between the complexity of machine learning models and their accuracy in emotion recognition? Do more complex models consistently outperform simpler ones, or are there instances where simpler models achieve comparable accuracy?
3. **Inherent Model Advantages and Model Comparison:** Are there certain machine learning classifiers that demonstrate inherent advantages in capturing the underlying patterns within EEG data for emotion recognition? How do these advantageous models compare to their counterparts in terms of accuracy and performance, and what insights can be gained from their differential performance?

These research questions guide the study's investigation into the performance, model complexity, and inherent strengths of machine learning classifiers in the context of emotion recognition using EEG data.

## 2.3. Novelty of This Study

The novelty of the above study lies in several key aspects:

**Emotion Recognition using EEG Data:** The study focuses on emotion recognition using EEG (electroencephalogram) data. EEG data provide a unique and direct insight into brain activity, allowing for the capture of neural responses associated with emotional states. This utilization of EEG data for emotion recognition is a novel approach, as it delves into the field of neuroscience and psychology to address a practical problem.

**Comprehensive Evaluation of Classifiers:** The study's novelty lies in its comprehensive evaluation of multiple machine learning classifiers for emotion recognition. While previous studies have explored emotion recognition using various classifiers, this study systematically compares a range of classifiers, including random forest, decision tree, logistic regression, support vector machine, stochastic gradient descent, and naive Bayes, among others. This thorough evaluation provides insights into the strengths and weaknesses of each classifier for this specific task.

**Identification of Top Performers:** The study identifies random forest and decision tree classifiers as the top performers for emotion recognition based on EEG data. While these classifiers have been used in various contexts, their application to EEG-based emotion recognition and their consistent high performance in this study contribute to the field's understanding of effective techniques for this novel task.

**Emphasis on Generalization:** The study's emphasis on the generalization ability of classifiers is novel. By showcasing that top-performing classifiers exhibit consistent accuracy across different validation set sizes, the study highlights the robustness of these models in real-world scenarios where varying amounts of data might be available.

**Insights into Model Complexity:** The study's exploration of the relationship between classifier performance and model complexity provides novel insights. It demonstrates that both complex (ensemble methods) and simpler (logistic regression) models can achieve commendable accuracy, suggesting that there is not a one-size-fits-all approach. This nuanced understanding is valuable for future model selection in emotion recognition tasks.

**Inherent Model Advantages:** The study's identification of certain classifiers, such as the nearest centroid and Gaussian naive Bayes, exhibiting superior accuracy performance compared to their counterparts, introduces a novel perspective. This hints at inherent advantages that specific models might have in understanding complex relationships within EEG data.

**Future Directions and Real-World Applicability:** The study's call for future research in exploring alternative algorithms, deep learning techniques for feature extraction, and comprehensive validation on diverse datasets highlights novel directions for advancing emotion recognition systems. Its focus on the practical applicability of these systems in real-world scenarios is a novel bridge between technical research and societal impact.

In summary, the novelty of the study stems from its application of EEG data for emotion recognition, its comprehensive evaluation of multiple classifiers, the identification of top-performing classifiers, insights into model complexity, discovery of inherent model advantages, and its emphasis on practical applicability and future directions. These elements collectively contribute to the study's originality and potential impact on the field of emotion recognition and mental health.

### 2.4. Significance of Our Work

The significance of this study lies in its exploration of emotion recognition using EEG (electroencephalogram) data and the evaluation of various machine learning classifiers for this task. The study reveals several important findings and implications:

**Classifier Performance:** The study comprehensively evaluates multiple machine learning classifiers for emotion recognition based on EEG data. It identifies random forest and decision tree classifiers as the top performers, demonstrating high training and validation accuracy. Logistic regression and support vector machine classifiers also perform well, achieving accurate predictions on unseen data instances. This performance hierarchy provides valuable insights into the efficacy of different classifiers for this task.

**Generalization Ability:** The study emphasizes the generalization prowess of the top-performing classifiers. These models maintain a wide range of accurate predictions across different validation set sizes, showcasing their robustness in dealing with unseen data. This generalization capability is crucial for real-world applications where new data instances might differ from the training data.

**Model Complexity:** The discussion highlights a trade-off between model complexity and performance. The top-performing classifiers are generally more complex, with decision trees and random forests being ensemble methods. In contrast, simpler models like logistic regression also exhibit commendable performance, indicating that a balance between model complexity and accuracy can be achieved.

**Inherent Model Advantage:** The study uncovers intriguing insights into specific classifiers, such as the nearest centroid and Gaussian naive Bayes, which outperform their counterparts. This suggests that certain classifiers might inherently possess advantages in understanding complex relationships within the provided EEG data. These findings prompt further investigation into the unique strengths of different algorithms.

**Future Research Directions:** The study acknowledges the dynamic nature of the machine learning field and identifies opportunities for future research. It suggests exploring alternative algorithms and the integration of multiple algorithms to enhance accuracy. Additionally, the study points towards the potential of deep learning techniques for feature extraction, which could lead to improved accuracy in deciphering complex EEG signals.

**Real-World Applicability:** The study's discussion underscores the need for comprehensive testing on diverse datasets to validate the robustness and versatility of the emotion recognition system in real-world scenarios. This real-world applicability is crucial for practical adoption in various domains where accurate emotion recognition can have significant societal impacts.

**Mental and Emotional Health:** The study's context aligns with the importance of mental and emotional health. Emotion recognition has the potential to contribute to

better understanding and addressing conditions like anxiety and depression. By adapting approaches based on emotional states, individuals' well-being can be enhanced.

**Advancements and Breakthroughs:** The study concludes by highlighting the potential of continued research in refining and advancing emotion recognition systems. Optimizing these systems could lead to breakthroughs in accuracy and efficacy, enabling their practical application in diverse domains.

In summary, this study's significance lies in its comprehensive analysis of machine learning classifiers for emotion recognition using EEG data. The findings contribute to the understanding of classifier performance, generalization, model complexity, and potential future research directions. Moreover, the study's focus on mental and emotional health underscores its societal relevance and potential positive impact.

## 3. Methodology

### 3.1. Dataset

Since EEG signals play a very prominent and interesting role in a brain–computer Interface (BCI), multiple publicly available datasets have been recorded over the years. Most of these datasets are either publicly accessible or available on request. For this study, we considered four different popular datasets for our experiments, as shown in Table 1. The first and the oldest dataset, DEAP [27], introduces a multimodal dataset designed to analyze human emotional states, including EEG and physiological data from 32 participants who watched music videos and provided ratings for various emotional aspects. Published in 2011, the dataset also incorporates frontal face videos for a subset of participants and utilizes a unique method for stimulus selection based on affective tags, video highlights, and online assessments. The SEED EEG dataset [28], provided by the BCMI laboratory and published first in 2013, includes EEG data along with vigilance datasets and has been widely utilized by over 2600 applications and 770 research institutions. It features 15 carefully selected Chinese film clips designed to elicit specific emotions, with each clip lasting approximately 4 minutes, accompanied by self-assessment periods and EEG data collection using a 62-channel ESI NeuroScan System and SMI eye-tracking glasses.

**Table 1.** Table depicting the accessible EEG datasets and their characteristics.

| Datasets | Reference | Year of Publication | # of Classes | # of People (Male + Female) | # of Minutes | Available From |
|---|---|---|---|---|---|---|
| EEG Brainwave Dataset: Feeling Emotions | Bird et al. [29] | 2019 | 3 (Negative, Positive, & Neutral) | 2 (1 + 1) | 9 | Kaggle ([30]) |
| DREAMER | Katsigiannis et al. [31] | 2017 | 3 (Violence, Arousal & Dominance) | 23 (14 + 9) | 59.7 | Zenodo ([31]) |
| SEED_EEG | Duan et al. [28] | 2013 | 3 (Negative, Positive, & Neutral) | 15 (7 + 8) | 960 | BCMI ([32]) |
| DEAP | Koelstra et al. [27] | 2011 | 3 (Arousal, Violence, & Like/Dislike) | 32 | 1280 | Prof. Ioannis Patras ([27]) |

In this investigation, we employed the EEG brainwave dataset, a publicly available dataset tailored for emotion recognition based on EEG signals. The number of classes in each dataset represents the number of output labels present in the dataset. The dataset exhibits balance, structured into three distinct categories: positive, negative, and neutral. Each classification harbors an equivalent number of samples, ensuring equilibrium. The main reason for choosing this dataset is its up-to-date nature, as it is the most current among the four datasets at our disposal. In addition, the three classes in this dataset had

(almost) equal number of classes, making this dataset balanced from the start. Furthermore, its higher reliability can be attributed to the use of cutting-edge technologies and an advanced EEG headset for data acquisition, which surpasses the equipment used in other contemporary datasets. As a result, this balanced dataset offers less noisy and more dependable data, despite its smaller size and fewer samples, making it a suitable choice for our analysis. The dataset, titled "emotions.csv", encompasses 2132 rows and 2549 columns, yielding a file size of 48.83 MB. This comprehensive dataset is readily available on the Kaggle platform. The dataset's attributes encompass a wide spectrum of features derived from EEG brainwave data, comprising statistics like mean, standard deviation, fast Fourier transformation outcomes, minimum and maximum values, eigenvalues, and entropy, among others. These attributes are encompassed within the initial 2548 columns. The categorical labels corresponding to the three emotion categories—positive, negative, and neutral—are placed within the final column. Each of these categories showcases a distinct count of values, with 708 values attributed to both positive and negative categories and 716 values aligned with the neutral category. Figure 1 graphically depicts the distribution pattern of the data labels within the dataset, offering a visual representation of the label distribution's characteristics.

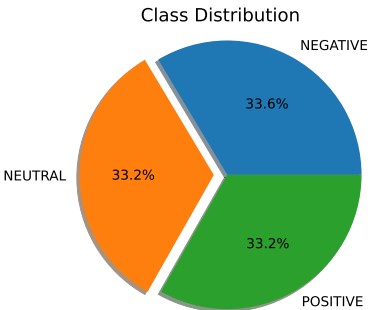

**Figure 1.** The percentage breakdown of positive, negative, and neutral responses to the EEG brainwave dataset from Kaggle as depicted by a pie chart.

The sample dataset utilized for the study is shown in Figure 2.

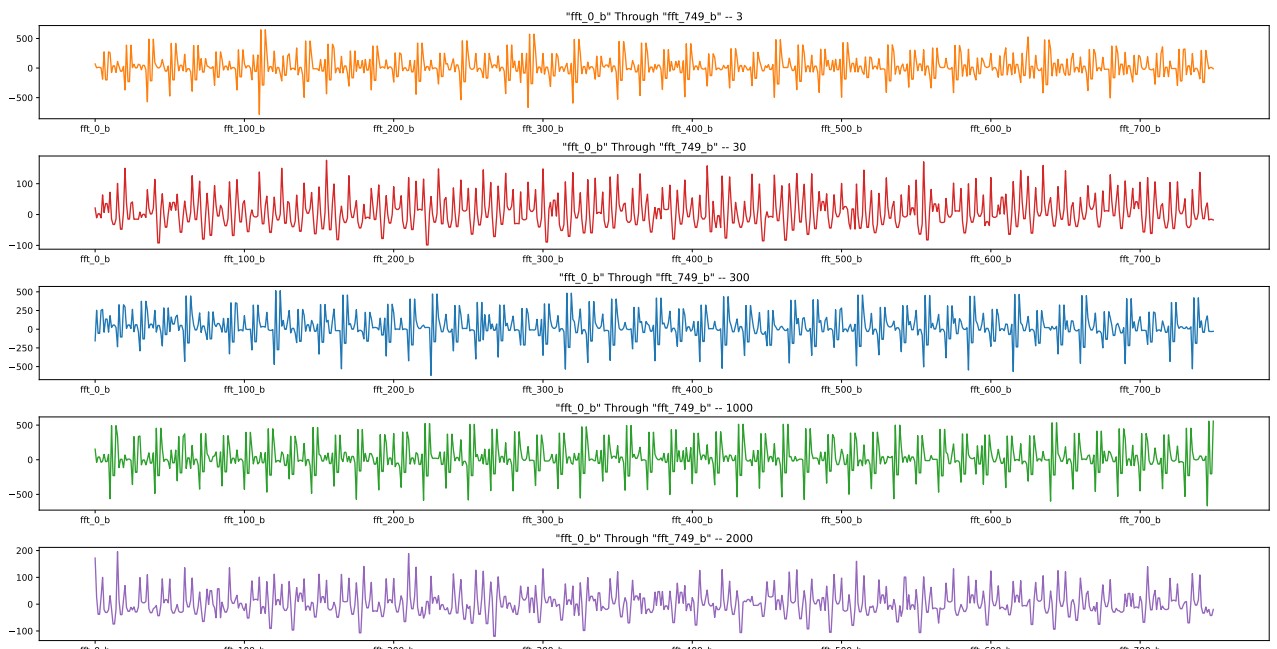

**Figure 2.** Pie chart showing the distribution of positive, negative, and neutral samples in the EEG brainwave dataset from Kaggle.

*3.2. Detailed Methodology*

The initial step involved preprocessing and validation of the dataset to ensure the absence of null values. The raw dataset consisted of 2549 features. We imported the raw data as DataFrame using pandas. This was followed by counting the number of labels for each of the three classes for the output variable, emotion. Then, we checked if a data point was missing in the whole dataset by checking for isnull() values in the dataframe, and the results indicated that no value in any cell was missing for this dataset. We then encoded the negative class by 0, neutral class by 1, and positive class by 2 for each output label and replaced the classes by their labels. After performing the basic sanity checks, e.g., via counting the unique values in the output label and counting the values in the output label, we dropped the label column from the data and formed the input matrix of size $2132 \times 2548$. Similarly, the output samples were saved in a vector, $y$, of size 2132. Then, we used the standard scaler and fitted and transformed the whole dataset using the scaler. Following this, the categorical nature of the label, with its three distinct values—positive, negative, and neutral—prompted the utilization of a label encoder to convert these categories into numeric counterparts. Subsequently, the dataset was partitioned into training and validation sets, employing four different ratios (90:10, 80:20, 70:30, and 60:40 as split ratio for training and validation) for a comprehensive range of experiments using the "train_test_split" function available in the sklearn.moedl_selection library.

The dataset underwent segmentation into training and validation subsets, serving as the foundation for a series of four distinct experiments. The first experiment allocated 90 percent of the dataset for training and 10 percent for validation. The second experiment entailed an 80:20 split, with 80 percent for training and 20 percent for validation. Subsequently, the third experiment adopted a ratio of 70 percent for training and 30 percent for validation, while the fourth and final experiment allocated 60 percent of the data for training and reserved 40 percent for validation.

Regarding machine learning classifiers, a diverse range of models were trained, encompassing a random forest classifier, decision tree classifier, logistic regression, support vector machine [33], stochastic gradient descent, nearest centroid, Gaussian naive Bayes, and Bernoulli naive Bayes [34,35]. Post classifier training, the process of hyperparameter tuning was conducted by employing GridSearchCV to identify optimal parameter configurations [36]. The methodology and workflow underlying our contributions are comprehensively outlined in Figure 3.

*3.3. Classifiers*

3.3.1. Random Forest

Random forest stands as a prominent supervised machine learning algorithm, employed across classification and regression domains. This technique constructs numerous decision trees from varied data samples and reaches the ultimate classification verdict through the consensus of these trees' majority votes. The algorithm's adaptability to handle both continuous and categorical variables renders it applicable to a wide spectrum of regression and classification challenges. Renowned for its prowess, random forest consistently surpasses alternative algorithms in diverse classification endeavors.

The following are some of the steps involved in inference or classification using the random forest classifier:

1. Bootstrapping: Create multiple samples from the original dataset by using bootstrapping.
2. Generation of Decision Trees: For each bootstrapped sample, build a decision tree where each tree continues to grow until all of its leaves contain only one class.
3. Feature Selection: Select a random subset of features for each split in each tree.
4. Classification: Classify new instances by using the majority vote of all the decision trees.
5. Regression: Compute the average predictions of all the trees for regression problems.
6. Pruning: Eliminate decision trees that do not have a significant impact on the final prediction to prevent overfitting.
7. Repetition: Repeat steps 1–6 multiple times to form an ensemble of trees.

8.  Final Prediction: Make the final prediction based on the majority vote of all the trees in the ensemble.

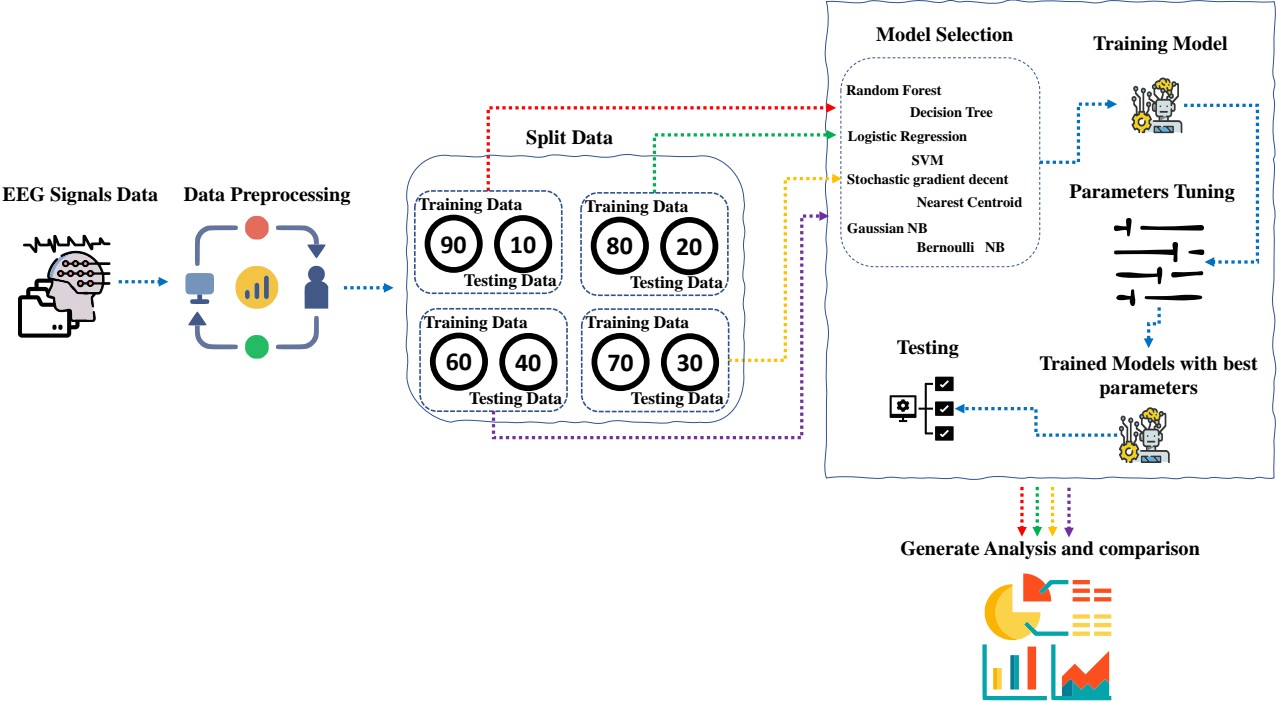

**Figure 3.** The overall working of the proposed solution.

Figure 4 illustrates the working of the random forest algorithm on the validation dataset.

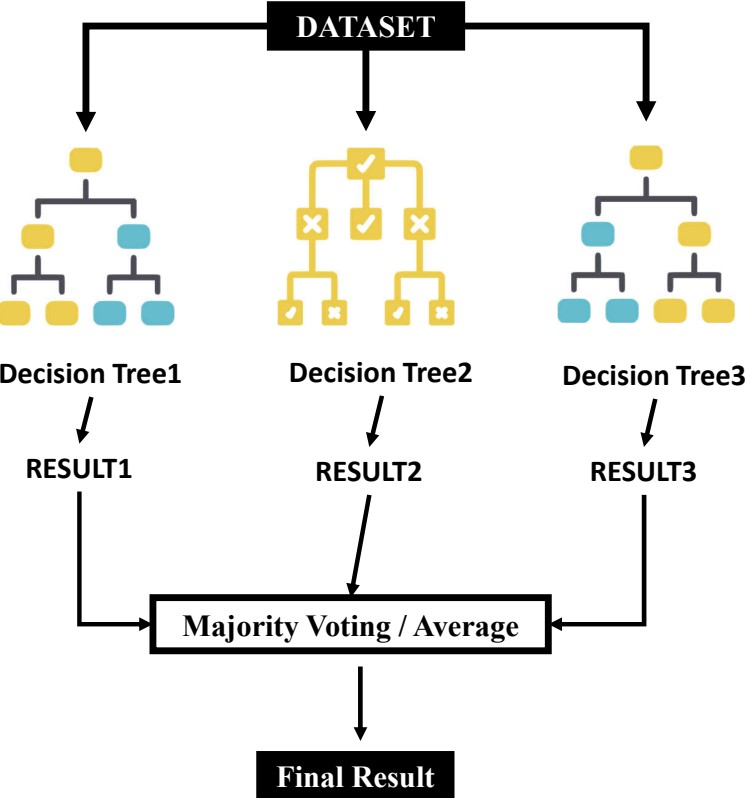

**Figure 4.** This diagram shows how the random forest classification method works.

### 3.3.2. Decision Tree

The decision tree classifier, a supervised machine learning approach for regression and classification endeavors, aspires to craft a predictive model for target variable class labels or values, leveraging decision rules gleaned from training data. In the creation of a decision tree, the process begins at the root node, contrasting the record's attribute against the root attribute. Based on this comparison, the algorithm traverses to the next node through the corresponding branch. This sequence endures until a leaf or terminal node is reached, ultimately delivering the record's classification. Each node in the tree embodies a test for a distinct attribute, with edges stemming from the node signifying possible responses to this test. This recursive process of testing and node traversal is reiterated for each subtree within the overarching tree.

### 3.3.3. Logistic Regression

Logistic regression, a supervised learning technique, evaluates data to foresee a categorical binary dependent variable's outcome. It comes into play when confronting binary classification quandaries, such as distinguishing between spam and non-spam emails. This approach employs a sigmoid function to gauge label likelihood, producing results within the range of 0 to 1 and generating an "S" curve graphically. An integral notion in logistic regression is the threshold value. In contrast, linear regression assumes a linear association between dependent and independent variables, while logistic regression confines its output to the 0–1 range. Figure 5 supplies an illustrative depiction of the logistic regression model's functioning.

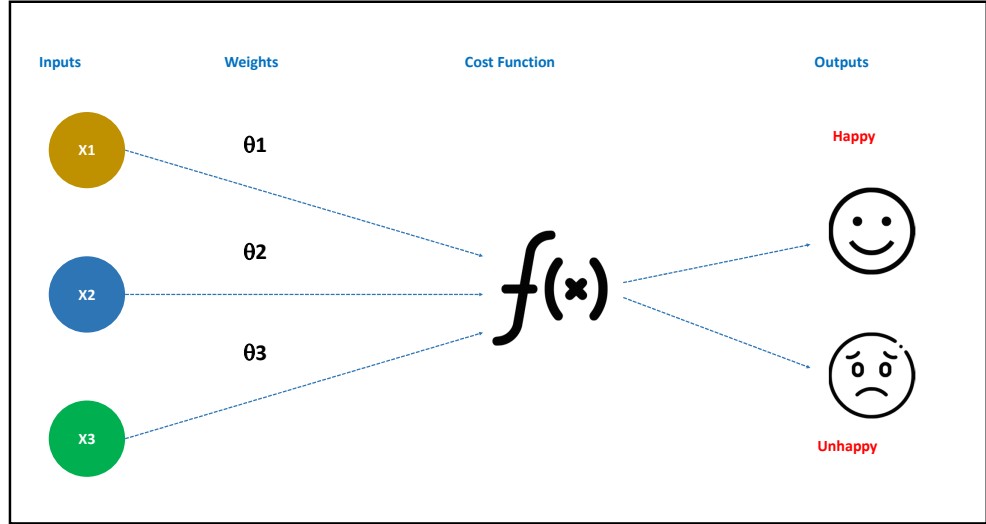

**Figure 5.** An example of how the logistic regression algorithm functions as a classification approach.

### 3.3.4. Support Vector Machine

Support vector machine (SVM) algorithm's primary objective is to discern a hyperplane within an N-dimensional space, aptly segregating distinct data points. From multiple plausible hyperplanes, SVM selects the one with the widest margin—signifying the separation distance between data point classes. This broader margin instills heightened classification robustness, fostering greater confidence in forecasting future data points. These hyperplanes operate as decision boundaries, aiding data point classification. Data points positioned on either side of the hyperplane can be categorized differently. The hyperplane's dimensionality corresponds to the feature count; for instance, in a two-feature scenario, it manifests as a line, expanding into a two-dimensional plane with three features. Visualizing beyond three features is intricate. Support vectors, pivotal data points proximate to the hyperplane, profoundly influence its orientation and location. Amplifying

classifier margin hinges on these support vectors; their absence would induce a hyperplane shift. A comprehensive grasp of these factors substantiates the SVM model's construction.

### 3.3.5. Stochastic Gradient Descent

Stochastic gradient descent (SGD) emerges as an effective technique to train linear classifiers and regressors using convex loss functions, such as linear support vector machines and logistic regression. The SGDClassifier presents a streamlined learning approach through stochastic gradient descent, accommodating diverse classification loss functions and penalties. This classifier readily handles multi-class classification, employing a "one versus all" (OVA) configuration by orchestrating multiple binary classifiers. For each of the K classes, a distinct binary classifier is trained. During the validation phase, confidence scores—reflecting the signed distances to the hyperplane—are computed for each classifier. The class boasting the highest confidence score is subsequently chosen. This concise framework encapsulates the fundamental workings of the SGDClassifier within a multi-class context.

### 3.3.6. Nearest Centroid

The nearest centroid classifier serves as a straightforward, yet effective machine learning approach tailored for classification tasks. While akin to the k-Nearest Neighbors (kNN) classifier, it features a more uncomplicated implementation. This classifier's core concept revolves around identifying the centroid (mean) of each class within the training data, subsequently employing this insight to label new data points. When presented with a novel data point, the nearest centroid classifier computes the distance between said point and the centroids of every class. Subsequently, the class with the closest centroid to the new data point is designated as its label. This iterative process extends to all new data points, thereby enabling the classifier to generate predictions for previously unseen data. This succinct framework encapsulates the foundational essence of the nearest centroid classifier in the context of classification tasks.

The nearest centroid classifier functions in the following manner.

1. The centroid for each target class is computed during training.
2. Say 'X' at any point after training. Distances are calculated between the point X and the centroid of each class.
3. From among all calculated distances, the shortest distance is chosen. The class is assigned to the centroid from which the given point is the shortest distance.

Figure 6 provides a schematic representation of how the nearest centroid model works.

### 3.3.7. Naive Bayes

The naive Bayes algorithm stands as a supervised machine learning method extensively used for classification tasks. It earns the "supervised" distinction due to its training employing a dataset encompassing both input features and categorical outcomes. The classifier's label "naive" stems from its presumption of feature independence, implying it disregards the interplay between input features. Nevertheless, this assumption may not universally hold.

### 3.4. Evaluation Metrics

The research employed established assessment criteria obtained from the confusion matrix, a common tool in classification analysis. The evaluation metrics were computed independently for both the training and validation datasets. In each case, accuracy was chosen as the primary metric due to its ability to strike a well-rounded equilibrium between precision, recall, false positives, and false negatives. Accuracy serves as a comprehensive measure that considers both the successful identification of positive instances and the avoidance of misclassifications. This deliberate choice of accuracy as the evaluation metric underscores the study's aim to capture the classifier's overall performance, ensuring that its predictions are both correct and minimally prone to errors. By assessing accuracy across

both the training and validation data, the study gains insight into the classifier's consistency in making accurate predictions, both in familiar scenarios and when faced with new, unseen data instances. This rigorous evaluation approach reinforces the study's commitment to comprehensively assessing the classifier's effectiveness and robustness.

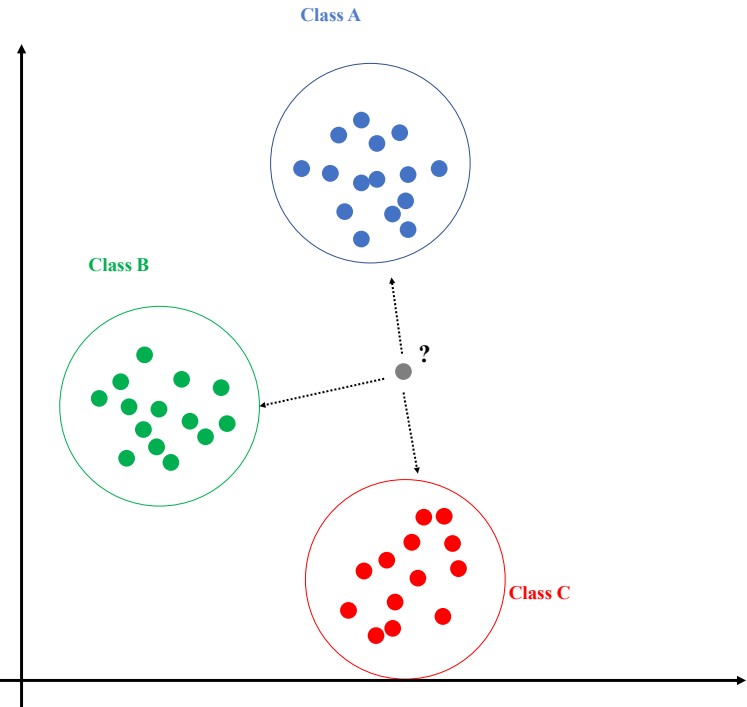

**Figure 6.** An example of how the kNN algorithm functions as a classification approach. Each color (blue, red and green) represents a unique class of data points and kNN algorithm is used to identify the group/cluster of each test point based on evidence from k nearest neighbor points.

### 3.5. Experimental Settings

In this study, we employed nine machine learning classifiers for the purpose of emotion recognition using EEG signals. The implementation was conducted using the Python programming language on the Google Colab platform. The ensemble of classifiers encompassed random forest, decision tree, logistic regression, support vector machine (SVM), stochastic gradient descent (SGD), nearest centroid, Gaussian naive Bayes (GNB), and Bernoulli naive Bayes (BNB). The dataset was partitioned into distinct training and validation sets, with the validation data constituting 10%, 20%, 30%, and 40% of the overall dataset.

### 4. Results

The training accuracy results for each classifier are as follows:

- For random forest, across training data proportions of 90%, 80%, 70%, and 60%, the accuracy was consistently 1.
- Similarly, decision tree exhibited identical results across all training data proportions.
- Logistic regression attained an accuracy of 1 across all training datasets.
- SVM achieved an accuracy of 0.98 for training data proportions of 90% and 80%, 0.97 for 60%, and 0.98 for 70%.
- SGD demonstrated an accuracy of 0.99 for 90%, 1 for 80% and 70%, and 0.99 for the 60% training data.
- Nearest centroid yielded an accuracy of 0.78 for 90%, 0.8 for 80% and 70%, and 0.79 for the 60% training data.

- Gaussian naive Bayes (GNB) showed an accuracy of 0.65 across all training data proportions.
- Bernoulli naive Bayes (BNB) achieved an accuracy of 0.83 for 90%, 0.84 for 80% and 70%, and 0.83 for the 60% training data.

Table 2 presents the training accuracy of the classifiers, providing comprehensive insight into the effectiveness of the data.

**Table 2.** Training accuracy for the classifiers on the EEG dataset.

| Classifiers | 90 Percent Data for Training | 80 Percent Data for Training | 70 Percent Data for Training | 60 Percent Data For Training |
|---|---|---|---|---|
| Random Forest | 1 | 1 | 1 | 1 |
| Decision Tree | 1 | 1 | 1 | 1 |
| Logistic Regression | 1 | 1 | 1 | 1 |
| SVM | 0.98 | 0.98 | 0.98 | 0.97 |
| SGD | 0.99 | 1 | 1 | 0.99 |
| Nearest Centroid | 0.78 | 0.8 | 0.8 | 0.79 |
| Gaussian NB | 0.65 | 0.65 | 0.65 | 0.65 |
| Bernoulli NB | 0.83 | 0.84 | 0.84 | 0.83 |

The validation accuracy outcomes for each classifier are outlined below:

- For random forest, the accuracy was 0.99 for the 10% validation data, and 0.98 for 20%, 30%, and 40% validation data.
- Decision tree demonstrated an accuracy of 0.97 for the 10% validation data, 0.95 for 20% validation data, 0.96 for 30% validation data, and 0.94 for 40% validation data.
- Logistic regression exhibited an accuracy of 0.97 for 10% validation data, and 0.96 for 20%, 30%, and 40% validation data.
- SVM achieved a consistent accuracy of 0.93 across all validation datasets.
- SGD's accuracy was 0.96 for the 10% validation data, 0.94 for 20% validation data, 0.95 for 30% validation data, and 0.93 for 40% validation data.
- Nearest centroid yielded an accuracy of 0.7 for the 10% validation data, 0.77 for 20% validation data, 0.78 for 30% validation data, and 0.77 for 40% validation data.
- Gaussian naive Bayes displayed an accuracy of 0.63 for the 10% validation data, and 0.65 for 20%, 30%, and 40% validation data.
- Bernoulli naive Bayes attained an accuracy of 0.81 for the 10% validation data, 0.82 for 20% testing data, 0.83 for 30% validation data, and 0.84 for 40% validation data.

Table 3 presents the validation accuracy of the classifiers, providing comprehensive insight into the performance of various models for the EEG dataset.

**Table 3.** Validation accuracy for the classifiers on the EEG dataset.

| Classifiers | 10 Percent Data for Validation | 20 Percent Data for Validation | 30 Percent Data for Validation | 40 Percent Data for Validation |
|---|---|---|---|---|
| Random Forest | 0.99 | 0.98 | 0.98 | 0.98 |
| Decision Tree | 0.97 | 0.95 | 0.96 | 0.94 |
| Logistic Regression | 0.97 | 0.96 | 0.96 | 0.96 |
| SVM | 0.93 | 0.94 | 0.94 | 0.94 |
| SGD | 0.96 | 0.94 | 0.95 | 0.93 |
| Nearest Centroid | 0.7 | 0.77 | 0.78 | 0.77 |
| Gaussian NB | 0.63 | 0.65 | 0.66 | 0.65 |
| Bernoulli NB | 0.81 | 0.82 | 0.83 | 0.84 |

## 5. Discussion

In the current study, we saw an application of machine learning algorithms for identifying the emotional patterns hidden in electroencephalogram (EEG) signals due to their

ability to uncover complex patterns and relationships within the data. The advantage of EEG signals is that they offer a direct window into the brain's electrical activity, providing valuable direct insight into emotional states. By employing machine learning algorithms, EEG data can be processed, and relevant features can be extracted, allowing the detection of subtle patterns associated with different emotions. These algorithms can learn from labeled datasets, associating specific EEG patterns with known emotional states, and subsequently, they can predict emotions in unlabeled data. Furthermore, machine learning models can adapt and generalize across a wide range of individuals, making emotion detection more personalized and robust. Fundamentally, the synergy between machine learning and EEG signals empowers emotion detection by translating intricate brain activity into actionable insights for various applications, including mental health monitoring, human–computer interaction, and affective computing.

Emotion detection from EEG signals holds significant potential in diverse applications. In mental health assessments, it can aid in diagnosing and monitoring conditions like depression and anxiety by objectively assessing emotional states. For user feedback, EEG-based emotion detection can enhance product development by gauging user reactions in real time, leading to more user-centric designs. In a human–computer interaction, it enables systems to adapt to users' emotional states, enhancing user experience and efficiency. In the context of social engineering, understanding emotional states from EEG signals can have ethical implications, as it may enable manipulative targeting of individuals. Therefore, while this technology offers promising benefits, its ethical use and potential risks must be carefully considered in each application domain.

Upon an in-depth analysis of the provided dataset, a clear pattern emerges, highlighting the random forest and decision tree classifiers as the frontrunners in terms of performance. These classifiers exhibit remarkable training accuracy levels of 1, signifying their adeptness in fitting the training data to perfection. Furthermore, their validation accuracy, which spans an impressive range from 0.94 to 0.99, underscores their generalization prowess when faced with unseen data. This wide spectrum of accurate predictions across different validation set sizes showcases the robustness of these models.

In the subsequent tier of performance, we find the logistic regression and support vector machine classifiers. Mirroring the top-performing models, these classifiers also achieve a training accuracy of 1, indicating their proficiency in capturing the intricate relationships within the training data. As for validation accuracy, their results vary between 0.93 and 0.97, demonstrating their ability to generalize and make accurate predictions on new, previously unseen data instances.

On the flip side, the remaining classifiers, including stochastic gradient descent, nearest centroid, Gaussian naive Bayes, and Bernoulli naive Bayes, manifest relatively lower training and validation accuracy rates. This divergence suggests that these classifiers might face challenges in capturing the underlying patterns within the data or in effectively generalizing their predictions to new instances.

Delving into the broader context of the emotion recognition experiment, the results collectively indicate that a majority of the employed classifiers achieved commendable accuracy scores. Particularly noteworthy are the performance levels of the random forest, decision tree, and logistic regression classifiers, which consistently demonstrated high accuracy rates across various validation scenarios. However, it is intriguing to observe that a subset of classifiers, specifically the nearest centroid and Gaussian naive Bayes, stood out with superior accuracy performance in comparison to their counterparts. This hints at the possibility that these models might have an inherent advantage in understanding the complex relationships within the provided data.

While certain methods employed in this investigation attain a training accuracy of 1, there may be a presumption of overfitting. Nevertheless, it is imperative to emphasize that the validation accuracy of such a method remains consistently high and does not exhibit a substantial decline, a characteristic indicative of overfitting. Consequently, it can be ascertained that overfitting is not evident for the method, even though it achieves a perfect

training accuracy overall. At the same time, shuffling validation data sizes also ensures that overfitting does not become a concern due to memorization or a relatively smaller size of dataset.

While this analysis highlights the strengths and areas for improvement within the classifiers, it is essential to acknowledge that the field of machine learning remains dynamic. The results of this experiment offer valuable insights, but there are still opportunities for refinement and further exploration. As this work progresses, it is likely that adjustments and enhancements can be made to harness the full potential of these classifiers for emotion recognition tasks. Thus, this study lays the groundwork for potential future endeavors aimed at enhancing the accuracy and robustness of emotion recognition systems.

To propel the efficacy of the emotion recognition experiment to greater heights, a strategic exploration of alternative machine learning algorithms, or the synergistic fusion of multiple algorithms, emerges as a potentially advantageous avenue. Diversifying the algorithmic approach could shed light on novel perspectives, potentially yielding improved accuracy rates in emotion identification tasks. This strategic diversification aligns with the multifaceted nature of emotion recognition, allowing for the harnessing of distinctive strengths inherent in different algorithms.

Expanding the frontier of accuracy enhancement involves delving into the realm of feature extraction methodologies. In this context, integrating more sophisticated techniques, particularly those grounded in deep learning paradigms, could potentially elevate the emotion recognition system's precision in deciphering complex EEG signals. The depth and adaptability of deep learning methodologies are well-suited for extracting intricate patterns and subtle nuances encapsulated within EEG data, potentially leading to enhanced emotion identification accuracy. Similarly, using Ensembl methods, e.g., XGBoost or some other gradient boosting method, might give more accurate classification results, as has been done in other studies on similar datasets. While the random forest did indicate significantly more accurate results, gradient boosting methods may be an interesting avenue to explore in the future.

Furthermore, subjecting the classifiers to comprehensive validation on more expansive and diverse datasets spanning various demographics holds immense promise. Such an endeavor transcends the boundaries of the controlled experimental setup, validating the system's robustness in real-world scenarios. This comprehensive validation would affirm not only the system's capacity to perform accurately across different situations but also its ability to generalize seamlessly across varying EEG signal inputs, attaining a level of versatility crucial for real-world applicability.

Although the current study showcases promising strides, it concurrently unveils a panorama of unexplored opportunities. As this research trajectory unfurls, numerous avenues for refinement and deeper investigation emerge. Engaging in further research with meticulous attention to various facets could potentially culminate in the optimization of the emotion recognition system anchored in EEG signals. This optimization, in turn, could wield the potential to usher in a new era of heightened accuracy and efficacy, empowering the practical application of the system across diverse domains. In essence, the pursuit of these directions represents a conscientious endeavor to maximize the capabilities of the emotion recognition system, thereby opening doors to multifaceted advancements and novel breakthroughs.

## 6. Conclusions

Human beings are inherently social creatures, reliant on interpersonal interactions for their well-being. Given the prevalence of anxiety and depression caused by diverse factors, maintaining vigilance over the mental and emotional health of those around us becomes imperative. Adapting our approach to individuals' emotional states can be profoundly advantageous, potentially alleviating their feelings of depression. This study centers on emotion detection using an EEG dataset. The paper demonstrates the effective utilization of a balanced dataset sourced from Kaggle. This readily labeled dataset is categorized

into three emotional states: positive, negative, and neutral. To classify these EEG signals, various machine learning techniques are employed. Notably, logistic regression yields the most promising outcomes, showcasing its potential for accurate classification of emotional states based on EEG data.

**Author Contributions:** Conceptualization, I.U.H. and R.H.A.; methodology, I.U.H.; software, I.U.H.; validation, R.H.A.; formal analysis, Z.u.A.; investigation, Z.u.A.; resources, T.A.K.; data curation, Z.u.A.; writing—original draft preparation, A.Z.I.; writing—review and editing, T.A.K.; visualization, T.A.K.; supervision, R.H.A.; project administration, T.A.K. All authors have read and agreed to the published version of the manuscript.

**Funding:** This research received no external funding.

**Institutional Review Board Statement:** Not applicable.

**Informed Consent Statement:** Not applicable.

**Data Availability Statement:** A publicly available dataset from Kaggle was analyzed in this study. Experiments were not carried out by the authors to collect these data, and hence, no ethical approval or consent was required. These data (EEG Brainwave Dataset: Feeling Emotions) can be found here: https://www.kaggle.com/datasets/birdy654/eeg-brainwave-dataset-feeling-emotions, accessed on 9 November 2023.

**Conflicts of Interest:** The authors declare no conflict of interest.

## Abbreviations

The following abbreviations are used in this manuscript:

| | |
|---|---|
| EEG | electroencephalogram |
| SVM | Support Vector Machine |
| SGD | Stochastic Gradient Descent |
| OVA | one versus all |
| kNN | k-Nearest Neighbors |
| BNB | Bernoulli Naive Bayes |
| GNB | Gaussian Naive Bayes |

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
