# Peer review of "Towards Effective Emotion Detection: A Comprehensive Machine Learning Approach on EEG Signals"

_biomedinformatics, doi:10.3390/biomedinformatics3040065_

Round 1

Reviewer 1 Report

Comments and Suggestions for Authors

please refer to the attachment .

Comments on the Quality of English Language

/

Author Response

Please see the attachment. We are grateful for your work on our manuscript and sincere and useful comments.

Reviewer 2 Report

Comments and Suggestions for Authors

The authors applied various classifiers to predict human emotions (positive, negative, and neutral) based on Electroencephalogram (EEG) signals from the Kaggle Emotion Detection dataset. This interdisciplinary research explored an innovative integration of machine learning methods and emotion detection. They compared the performance of these classifiers under different train-test split ratios, which gave some hints for potential feature applications. 

Comments and suggestions to the authors:

  1. This paper basically displays results for hyper-parameter tuning on several sophisticated algorithms on an open-source dataset. Considering this work is an application of emotion detection, it would be better to have an in-depth understanding of the relationship between these machine-learning methods, EEG signals, and emotion detection, especially why and how could these machine learning methods and EEG signals help emotion detection. 

  2. I have a concern on the potentially insufficient data. It would be great if the authors could have more adequate discussions about potential overfitting issues. For example, why did the training accuracy reach 1 in some scenarios? Did the authors try to shuffle the training, validation, and test data? Did the authors explore some data augmentation methods if no additional data could be found?

  3. The authors had an interesting argument that a balanced dataset is a more accurate representation of real-world emotion recognition scenarios. This argument could be more concrete if the authors could provide more evidence, such as statistics on emotional states, etc. In addition, the labels (i.e. the positive, neutral, and negative emotions) are not clearly defined in this paper. 

  4. There are misconceptions on some terms such as training, validation, and test from a statistical or machine learning point of view. 

  5. It would be better if the authors could provide more details for many important information. For example, more details of the dataset such as how the label is defined, what are the features, etc. It would also be great if the authors could explain more background instead of just some vague introductions on well-known general information, for example, why this dataset is used in this study, as well as a comparison between this study and other similar studies on the same or similar datasets. 

  6. I have a minor suggestion on the models, the authors tried random forest and decision trees. I’m wondering did the authors try some gradient boosting methods such as XGBoost?

Author Response

Thank you for your time, dedication, and useful comments and suggestions in helping us improve the quality of our manuscript. We are grateful to you.

Round 2

Reviewer 1 Report

Comments and Suggestions for Authors

i do not have any

Author Response

Dear Reviewer,

Thank you for taking out your time in the first round and providing us with useful comments. Your reviews and comments improved the quality of our work substantially. 

Kind Regards,

Prof. Dr. Raja Hashim Ali and team.

Reviewer 2 Report

Comments and Suggestions for Authors

Thanks for the authors' diligent work. The quality of this paper was significantly improved. The authors have addressed all the comments and suggestions. There are just some very minor typos that need proofreading, for example, "Table ??" at the end of line 308. 

Author Response

Dear Reviewer,

Thank you for your help, comments, and suggestions, and for taking out time to go in depth with the review. We appreciate your effort in helping us improve the quality of the manuscript. We have now incorporated the mentioned corrections and resolved a few other issues as well. Thank you.

Kind Regards,

Prof. Dr. Raja Hashim Ali and team.